# Towards precision apiculture: Traditional and technological insect monitoring methods in strawberry and raspberry crop polytunnels tell different pollination stories

**Scarlett R. Howard**[1]*, **Malika Nisal Ratnayake**[2], **Adrian G. Dyer**[3,4], **Jair E. Garcia**[3], **Alan Dorin**[2]*

**1** Centre for Integrative Ecology, School of Life and Environmental Sciences, Deakin University, Burwood, Victoria, Australia, **2** Faculty of Information Technology, Monash University, Clayton, Victoria, Australia, **3** School of Media and Communications, RMIT University, Melbourne, Victoria, Australia, **4** Department of Physiology, Monash University, Clayton, Victoria, Australia

* s.howard@deakin.edu.au (SRH); alan.dorin@monash.edu (AD)

**Data Availability Statement:** The data are available in Monash Bridges: https://doi.org/10.26180/14531892.

## Abstract

Over one third of crops are animal pollinated, with insects being the largest group. In some crops, including strawberries, fruit yield, weight, quality, aesthetics and shelf life increase with insect pollination. Many crops are protected from extreme weather in polytunnels, but the impacts of polytunnels on insects are poorly understood. Polytunnels could reduce pollination services, especially if insects have access issues. Here we examine the distribution and activity of honeybees and non-honeybee wild insects on a commercial fruit farm. We evaluated whether insect distributions are impacted by flower type (strawberry; raspberry; weed), or distance from polytunnel edges. We compared passive pan-trapping and active quadrat observations to establish their suitability for monitoring insect distribution and behaviour on a farm. To understand the relative value of honeybees compared to other insects for strawberry pollination, the primary crop at the site, we enhanced our observations with video data analysed using insect tracking software to document the time spent by insects on flowers. The results show honeybees strongly prefer raspberry and weed flowers over strawberry flowers and that location within the polytunnel impacts insect distributions. Consistent with recent studies, we also show that pan-traps are ineffective to sample honeybee numbers. While the pan-traps and quadrat observations tend to suggest that investment in managed honeybees for strawberry pollination might be ineffective due to consistent low numbers within the crop, the camera data provides contrary evidence. Although honeybees were relatively scarce among strawberry crops, camera data shows they spent more time visiting flowers than other insects. Our results demonstrate that a commercial fruit farm is a complex ecosystem influencing pollinator diversity and abundance through a range of factors. We show that monitoring methods may differ in their valuation of relative contributions of insects to crop pollination.

**Funding:** SRH acknowledges Monash University and the Alfred Deakin Postdoctoral Fellowship. AGD and AD were supported by the Australian Research Council Discovery Projects grant DP160100161. AD acknowledges the support of a Monash-Bosch AgTech Launchpad primer grant for this research. The funders had no role in study design, data collection and analysis, decision to publish, or preparation of the manuscript.

**Competing interests:** Adrian G. Dyer is an Academic Editor for PLOS ONE. This does not alter our adherence to PLOS ONE policies on sharing data and materials.

## Introduction

Over one third of food production from crops relies on animal pollination, with up to 70% of major crop species needing it to some extent [1]. Bees, flies, beetles, moths, butterflies, wasps, ants, birds, bats, and other animals all contribute to angiosperm and global crop pollination [2]. In fact, 87.5% of flowering plants use animals in pollen transfer [3] with about 300,000 animal species involved [4]. Insects are the principal pollinator of agricultural plants that require animal visitations [1,5,6]. The estimated worldwide annual value of crop pollination by insects is US$235 –US$577 billion [6]. Since bees are emphasised as one of the most important and effective pollinators of crops and other plants, and honeybees and bumblebees can both be managed at an industrial scale, they have long been deliberately introduced and managed on farms [7]. In recent times it has become appreciated that non-bees and wild insects are also very important for some crops [8]. Non-bee insects perform approximately 25–50% of total flower visits globally, make up 39% of visits to crop flowers [2], and fruit sets tend to increase with non-bee insect visitation [2,8]. However, in contrast to bumblebee and honeybee pollination, alternative insect pollinators, including native bees, are less frequently managed. In Australia, there are some exceptions to this including stingless bees [9] such as *Tetragonula carbonaria* [10], and other native bee genera and families, including *Xylocopa*, *Amegilla*, and Megachilidae spp. [11–17]. Despite this work, at present there is a relative dearth of empirical evidence on the contribution of alternative insect pollinators to Australian agriculture.

Crop yield, quality, shelf life, and commercial value of strawberries (*Fragaria* sp.) are shown to increase with bee pollination [18–21]. Between 6 and 16 visits from honeybees are required for full pollination of a strawberry flower [22–24] and 12–25 honeybee hives per hectare are advised [22,24–26]. It has previously been reported that honeybees are often not very attracted to the nectar and pollen of strawberry flowers [22,24,25]. Conversely, raspberry (*Rubus* sp.) flowers appear to be attractive to honeybees due to a relatively high quantity of nectar [24,27]. Since raspberry flowers are attractive, fewer hives are needed for pollination of these flowers when compared to strawberries, with data indicating only between 0.5 and 2.5 hives per hectare are required for raspberries [22]. In addition, raspberry plants benefit from insect diversity, producing more and larger fruit when a variety of bees are involved in pollination [24].

As extreme weather events escalate, crops are increasingly planted within polytunnels of glass or semi-transparent plastic [28,29]. Polytunnels aid in the manipulation of the microenvironment, usually for the benefit of crop plants. Research by Hall et al. (2019) [30] found that the centres of polytunnels experience higher temperatures and reduced wind speed, but also lower abundances and fewer visits from stingless bees and honeybees, than the edges [30]. Thus, it appears that polytunnels may impact pollinator access, diversity, abundance, distribution, and plant reproduction/crop yield, raising questions about how to monitor and improve this increasingly common and important horticultural infrastructure. We examined the distribution of honeybees versus non-honeybee insects on a commercial strawberry and raspberry farm in South Eastern Australia to assist in understanding the value of investment in managed honeybee hives as compared to other insects available in the environment. Insect data was collected via passive and active survey methods–pan-traps and quadrat observations respectively. We examined the effects of flower type and polytunnel length on insect counts. In addition, video recordings were analysed using machine-learning and computer vision techniques to calculate the extent of flower visitation by honeybees versus other insects, as opposed to simply documenting the presence of these insects among polytunnel crops using the pan-traps and quadrat observations.

## Materials and methods

### General procedure

The counts and distributions of insects on a commercial strawberry and raspberry farm were measured during the flowering season in 2020. Data was collected using three methods: 1) researchers observed and counted insects (honeybees vs. other insects) in a 1 m x 2.2 m quadrat for 3-minute periods; 2) groups of blue, yellow and white pan-traps were set out over 24-hour periods; 3) a high-resolution camera was placed in the strawberry crop to capture video footage in a four-hour block, with the aim to enhance study depth by revealing supplementary detail of insect-flower interactions.

Pan-traps and quadrat observations counted insects to allow us to 1) compare insect counts between strawberry fields, raspberry fields, and unmanaged weed areas, and 2) determine insect counts along polytunnel lengths in the strawberry and raspberry fields. We also compared quadrat and pan-trap collection methods to understand how they might differently reflect insect distributions in each region.

### Farm and collection information

We collected data over a focussed period of five days (9th– 13th March 2020) at Sunny Ridge farm, an active commercial strawberry and raspberry grower in Boneo, Victoria, Australia (Fig 1). Victoria is a temperate region which only allows some species to live in Victoria all year around. Early March is within the peak period for pollination service requirements for strawberries in Victoria [22,25] and so we selected this window for our study in concert with the needs of the farm manager to provide a snapshot of consistent peak activity (rather than a longer study which would instead have explored seasonal changes in pollinator activity, something that was not our intention). During this time insecticide was not sprayed in the areas of data collection. Fungicide was sprayed but was not expected by Sunny Ridge Farm

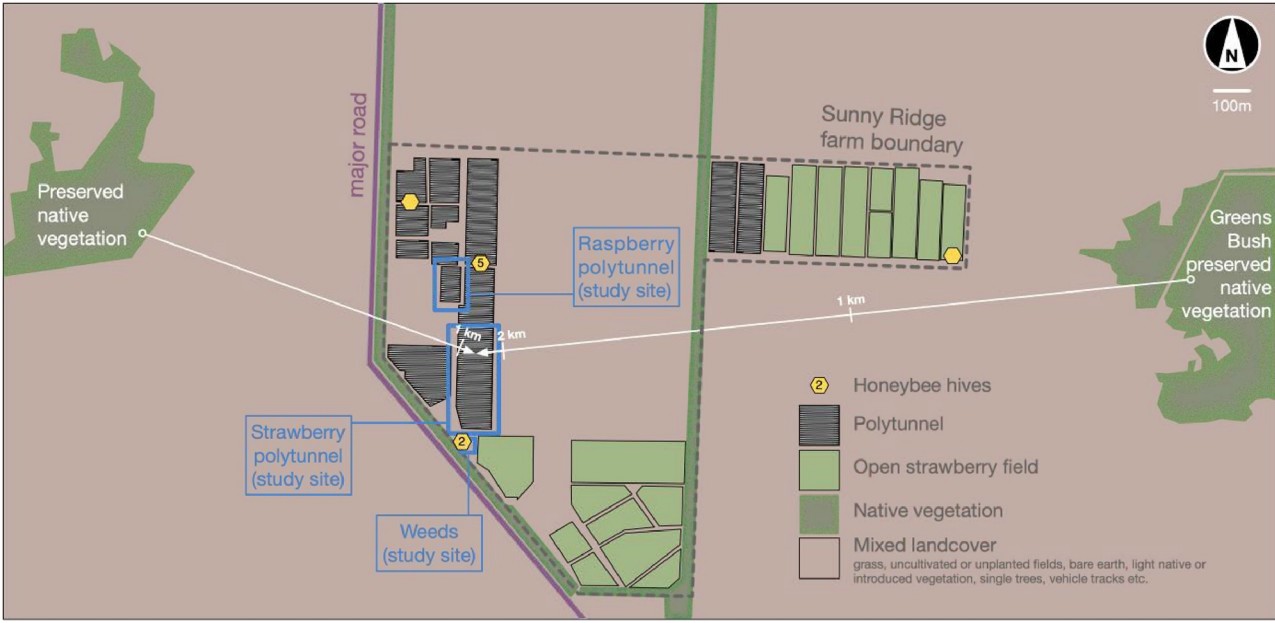

**Fig 1. Satellite image of Sunny Ridge farm 38˚25'25.2"S 144˚53'33.8"E (perimeter indicated by dotted grey line).** Managed honeybee hives (yellow hexagons) with the number of hives at locations close to study sites indicated. Raspberry/strawberry polytunnels and the weed location used in the study are marked. Nearby areas of native bushland are indicated along with their ranges from the strawberry polytunnel for reference.

management to directly impact insect abundance during the observation period (although, see [31]). Belt® 480 SC Insecticide was sprayed on the 6th– 7th March 2020 in the area and is applied to crops approximately every 1–2 weeks to control for pest species, such as thrips. The size of the farm is approximately 90 hectares (Fig 1). Sunny Ridge reports (personal communication, Oct. 2020) up to 20–30% of strawberry crop waste (i.e. fruit unsuitable for commercial sale) at some points in the season, with an average of around 10% overall. This is thought to be at least partially due to mis-pollination or a lack of pollination. Strawberry varieties where we monitored insects were Cabrillo [32] and Albion [33].

The area surrounding the farm consists of other agriculture and rural development, remnant native habitat, and a large section of preserved bushland (Greens Bush) within a 2 km radius of the study location (Fig 1). A major road runs along the west/south-west boundary of the farm.

Polytunnel strawberry fields varied from 78–96 m long and 8–9 m wide with 4 rows of strawberry plants per polytunnel. Raspberry polytunnels were 50 m long and 8–9 m wide with 4 rows of raspberry plants per polytunnel. Polytunnel roofing material was translucent LDPE (diffusing) plastic.

## Locations

Study locations (Fig 1) included strawberry and raspberry crops within polytunnels, and (uncovered) weeds on the farm. At the time of the study the property included four sets of managed honeybee hives (2–10 hives per site) as shown in Fig 1.

## Collection methods

According to a recent evaluation of insect capture techniques on Australian insects [34], active methods (such as observation) exceed other passive methods at collecting bee abundance data but are less effective if finer taxonomic categorisation is necessary [34,35]. We used both active (observation in quadrats) and passive (pan-traps) methods to collect insect abundance data over five days. This was supplemented with video observations on one day to enable detailed study of insect-flower interactions.

**Quadrat observations.** Quadrat observations consisted of two researchers examining a half each of a 1 m × 2.2 m area of crops or weeds for three minutes and recording all insect sightings on flowers, or in the air, whilst avoiding immediate double-counting. Observations were made at multiple locations in the strawberry and raspberry polytunnels, and the weed sites, between 10.00am and 3.00pm each day. See more details for each experiment below.

**Pan-traps.** Blue, yellow, and white pan-traps [36] were filled with 250 ml of water and a drop of unscented washing liquid [36,37]. They were placed in groups, with each trap at least 1 m from the others in its group within the strawberry and raspberry polytunnels, and weeds (Fig 2). Pan-traps were set for 24 hours at a time. After this, the insects collected in the traps were stored in 70% ethanol and the pan-traps were reset. See more details for each experiment below.

**Camera observations.** The camera system consisted of a Raspberry Pi Camera Module V2 connected to a Raspberry Pi 3 model B+ board and powered using a portable power bank. The camera was set up 700 mm above strawberry flowers with a field of view of ~45˚ × 25˚. Dimensions of the area covered by the camera were 600 *mm* × 340 *m*. Videos were recorded with a resolution of 1920 × 1080 at 30 frames per second (fps) in the period 10.00am am– 3.00pm. H264 codec was used for video recordings and later converted to MP4 format for analysis.

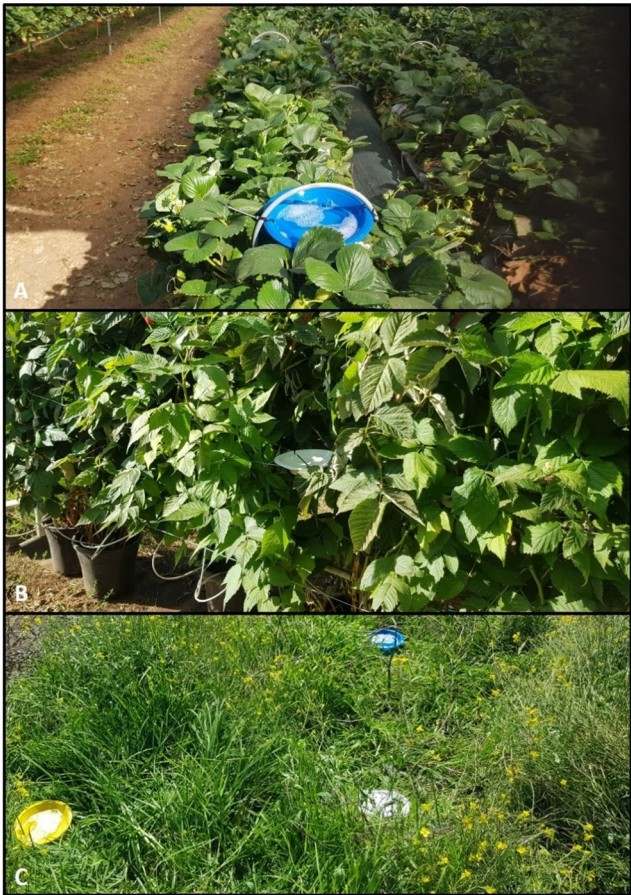

**Fig 2. Pan-trap locations.** (A) Strawberry polytunnel, (B) raspberry polytunnel, (C) open weed area. A blue, yellow, and white pan-trap were placed in each location to allow for known impacts of colour on insect counts of different species [36].

Movement tracks of honeybees were extracted from recorded videos (Fig 3) using automated tracking software [38]. This software uses a Hybrid Detection and Tracking (HyDaT) algorithm, comprising of foreground-background segmentation and deep learning-based detection techniques to identify and track honeybees in complex dynamic environments. Initially, the deep learning-based detection model is trained with honeybee images. It then

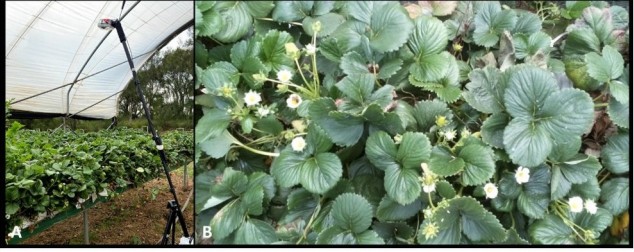

**Fig 3. Video-capture detail.** (A) Raspberry Pi and camera setup in the strawberry fields. (B) A frame from the recorded video.

identifies when a honeybee enters the field of view of the camera as recorded in the video. Once a honeybee has been accurately identified, depending on the extent of variation in the background, the algorithm intelligently switches between background-foreground segmentation and deep learning-based detection to track the honeybee until it leaves the field of view.

Extracted data were filtered to remove tracks less than one second in duration. Remaining honeybee tracks were used to calculate recorded foraging times and time spent specifically on flowers. Data related to other species of insect pollinator were extracted through human observation of videos. A validation experiment comparing human observations of the video footage and software analysis is described below.

We conducted two sets of statistical analysis to (1) compare the number of flowers visited and (2) the time spent on flowers by honeybees and other insects. We analysed flower visitation counts with a generalised linear model assuming a negative binomial distribution for the response variable using the "MASS" [39] package within the R environment for statistical analysis. We considered all the insects recorded within the field of view of the camera for this analysis, and insects which did not visit any flower were considered to have zero number of visits. We then compared the average time each insect type (honeybee / other insects) spent foraging on flowers using a Mann-Whitney U test as the time variable was not normally distributed. Only the insects which visited flowers were considered for the calculation of means.

**Validation study.** A study was conducted to validate foraging data collected with the automated software [38] on honeybees. For this, foraging time data (time spent on flowers) for 10 randomly selected honeybee tracks were compared against human observations of the video footage and percentage errors were recorded.

## Comparison analysis

To compare the collection methods of quadrat observations (active) and pan-traps (passive), we analysed insect count data with a generalized linear mixed effect model (GLMM) with a Poisson distribution using the "glmer" package within the R environment for statistical analysis [40]. We fitted a full model with collection type as a fixed predictor with two levels (quadrat observation; pan-traps) and insect counts (either "honeybees" or "other insects") as the dependant variable. Location of the observation/trap was included as a random factor to account for repeated measurements at each location. We used only the middle of the polytunnels for this comparison to avoid edge effects of polytunnels.

**Flower type as a predictor of insect counts.** We aimed to determine if managed honeybee and other insect counts differed between strawberries, raspberries, and weed sites (Fig 4). We compared insect counts between the strawberry polytunnels, raspberry polytunnels, and the weed sites on the farm using quadrat observations and pan-traps (described above). Data collected using observations and pan-traps were from the middle of the strawberry and raspberry polytunnels to avoid polytunnel edge effects. For the quadrats, there were N = 30 observations in strawberries, N = 9 observations in raspberries, N = 6 observations in weeds, each of three minutes duration. For insect counts in pan-traps, three pan-traps (blue, yellow, white) were set out at each location. The number of locations were N = 12 pan-taps for strawberries, N = 4 pan-traps for raspberries, N = 3 pan-traps for weeds, each of 24 hours duration.

To test for the effect of flower type on insect counts during quadrat observations and pan-trapping, we analysed count data with a generalized linear mixed effect model (GLMM) with a Poisson distribution using the "glmer" package within the R environment for statistical analysis [40]. We fitted a full model with flower type as a predictor with three levels (strawberries;

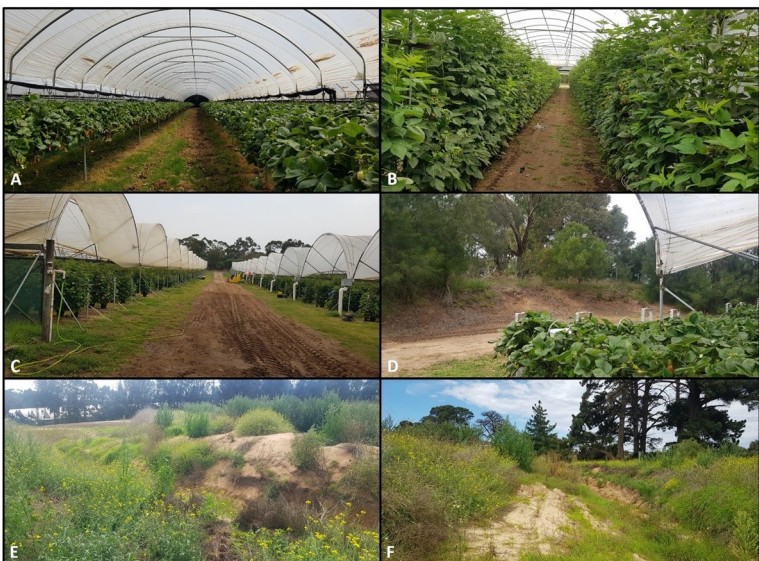

**Fig 4. Farm locations.** (A) Strawberry polytunnel, (B) raspberry polytunnel, (C) polytunnel edges near the farm road, (D) strawberry polytunnel near remnant habitat, (E-F) weed locations.

raspberries; weeds) and location of the quadrats or pan-traps as a random factor to account for repeated measurements collected on each location.

**Polytunnel location as predictor of insect counts in strawberry and raspberry crops.**
We aimed to determine if honeybee and other insect numbers differed between the edge and middle of the polytunnels as observed in previous studies [30] (Fig 5). We compared insect counts at the edges of the polytunnels (Strawberries: 1–12 meters from the edge; Raspberries: 1–5 meters from the edge) with the middle of the polytunnels (Strawberries: 35–52 meters from the edge; Raspberries: 22–26 meters).

In the strawberry polytunnels, we examined two locations along polytunnels: i) edges and, ii) the middle. There were $N = 30$ quadrat observations for the middle of the strawberry polytunnels and $N = 60$ observations for the edges ($N = 30$ per edge east / west). (Figs 4 and 5A). At each location three pan-traps (blue, yellow, white) were set out. $N = 12$ observations were made in the middle and $N = 24$ were made at the polytunnel edges ($N = 12$ per edge east / west) (Figs 4 and 5A).

For the strawberry polytunnels, a GLM model was fitted with polytunnel location as a predictor with two levels in the strawberry fields (edge of polytunnel; middle of polytunnel) with insect count as the dependent variable. Quadrat location between strawberry field polytunnel locations was a random factor.

In the raspberry polytunnels, we examined two locations along the polytunnel, the: i) edge; and ii) the middle. There were $N = 9$ quadrat observations for raspberries at the edge and $N = 9$ for raspberries in the middle of the polytunnel (Fig 5B). Three pan-traps (blue, yellow, white) were set out at each location. The number of observations was $N = 4$ for raspberries at the edge and $N = 4$ for raspberries in the middle of the polytunnels.

For the raspberry polytunnels, a GLM model was fitted with polytunnel location as a predictor with two levels in the raspberry fields (edge of polytunnel; middle of polytunnel), and insect count as the dependent variable. Quadrat location between raspberry polytunnel locations was a random factor.

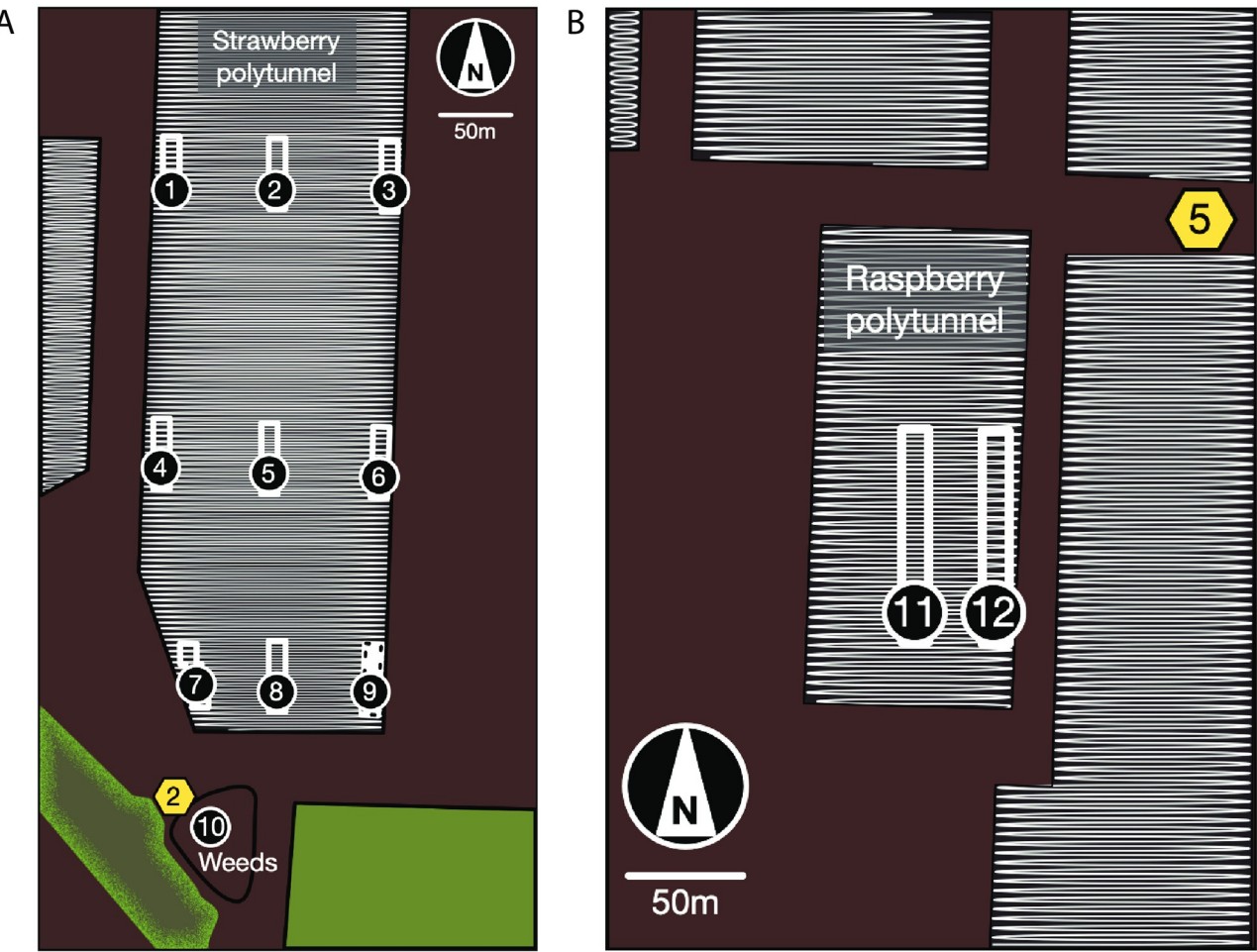

**Fig 5. Detailed plan of study site.** (A) Strawberry polytunnels and (B) raspberry polytunnels (refer to Fig 1 for site overview). Numbers/rectangles indicate locations of insect quadrat observations and pan-traps under the polytunnels #1–9, #11–12, and #10 in the weeds. The white-black dashed rectangle (location #9) additionally shows the location of the camera observations to record detailed insect- strawberry flower interactions. Managed honeybee hives (yellow hexagons) with the number of hives at locations close to study sites are indicated.

## Results

### Flower type as a predictor of insect counts

**Quadrat observations.** We used a GLMM to determine whether honeybee and other insect counts differed between strawberry polytunnels, raspberry polytunnels, and the weed sites on the farm when assessed using quadrat observations. Our model showed that flower type had a significant effect on honeybee counts ($z = 13.606$; $P < 0.001$) but no significant effect was apparent on the counts of other insects ($z = 0.481$; $P = 0.631$; Table 1; Fig 6).

**Table 1. Average insect count per quadrat along the polytunnels and in the middle of the uncovered weed with the mean and standard error of the mean (SEM) shown in brackets.**

| Polytunnel location | Strawberry polytunnels | | Raspberry polytunnels | | Weed area | |
|---|---|---|---|---|---|---|
| | Honeybees | Other insects | Honeybees | Other insects | Honeybees | Other insects |
| Edges | 0.75 (0.13) | 2.2 (0.25) | 16.67 (1.88) | 2.33 (0.53) | N/A | N/A |
| Middle | 0.33 (0.09) | 2.2 (0.37) | 17.89 (2.26) | 4.22 (1.37) | 29.83 (8.40) | 11.67 (2.88) |

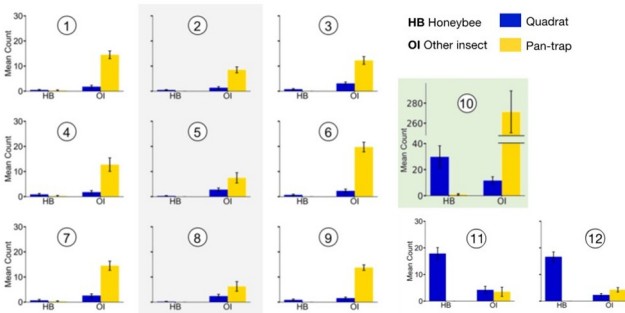

**Fig 6. Quadrat observations (blue) and pan-trap counts (yellow) of honeybees (HB) and other insects (OI).**
Strawberry polytunnels (1–9), weeds (10–note axis scale), raspberry polytunnels (11–12)–see Fig 5 for polytunnel
layout. Grey shading indicates the middle of the strawberry polytunnels, green shading indicates the weed area. Error
bars represent the standard error of the mean.

**Pan-traps.** Flower type had a significant effect on non-honeybee insect counts ($z$ = -4.659; P < 0.001; Table 2; Fig 6). However, too few honeybees were captured in pan-traps (see Discussion) to analyse this data meaningfully (Table 2).

## Location within polytunnel as predictor of insect counts in strawberry and raspberry crops

**Quadrat observations.** Our model showed that in strawberry polytunnels there was a significant difference between the middle of the polytunnel and its edges ($z$ = 2.320; P = 0.020; Table 1), as there were more honeybees observed near the edges than the middle of the polytunnel. Location within the polytunnel did not impact non-honeybee insect counts during quadrat observations in strawberry fields when comparing the middle to the edges ($z$ = 0.003; P = 0.998; Table 1).

In the raspberry fields, location within the polytunnel did not impact honeybee counts ($z$ = 0.62; P = 0.533; Table 1) but did have a significant effect on the numbers of other insects counted during quadrat observations in raspberry fields ($z$ = 2.181; P = 0.029; Table 1; Fig 6).

**Pan-traps.** Our model showed that location within the polytunnel significantly impacted non-honeybee insect counts in pan-traps in strawberry fields ($z$ = 5.696; P < 0.001; Table 2; Fig 6). As there were very low numbers of honeybees found in any area in pan-traps, statistical analysis was not possible for the numbers of honeybees along polytunnels in the strawberry crop. It is important to note that the honeybee count was also very low in pan-traps placed outdoors at the weed site; whilst pan-traps performed well at capturing other insects (Fig 6).

No honeybees were captured in pan-traps in the raspberry crop at all, and so there is no pan-trap data about them for analysis. Location within the raspberry polytunnel did not have a significant effect on the counts of non-honeybee insects in pan-traps ($z$ = -0.538; P = 0.591; Table 2; Fig 6).

**Table 2. Average insect count per pan-trap location along the polytunnels with the mean and standard error of the mean (SEM) shown in brackets.**

| Polytunnel location | Strawberry polytunnels | | Raspberry polytunnels | | Weed area | |
|---|---|---|---|---|---|---|
| | Honeybees | Other insects | Honeybees | Other insects | Honeybees | Other insects |
| **Edges** | 0.13 (0.07) | 14.58 (0.84) | 0.00 (0.00) | 4.25 (0.26) | N/A | N/A |
| **Middle** | 0.00 (0.00) | 7.42 (0.96) | 0.00 (0.00) | 3.50 (1.76) | 1.00 (0.58) | 271.33 (20.92) |

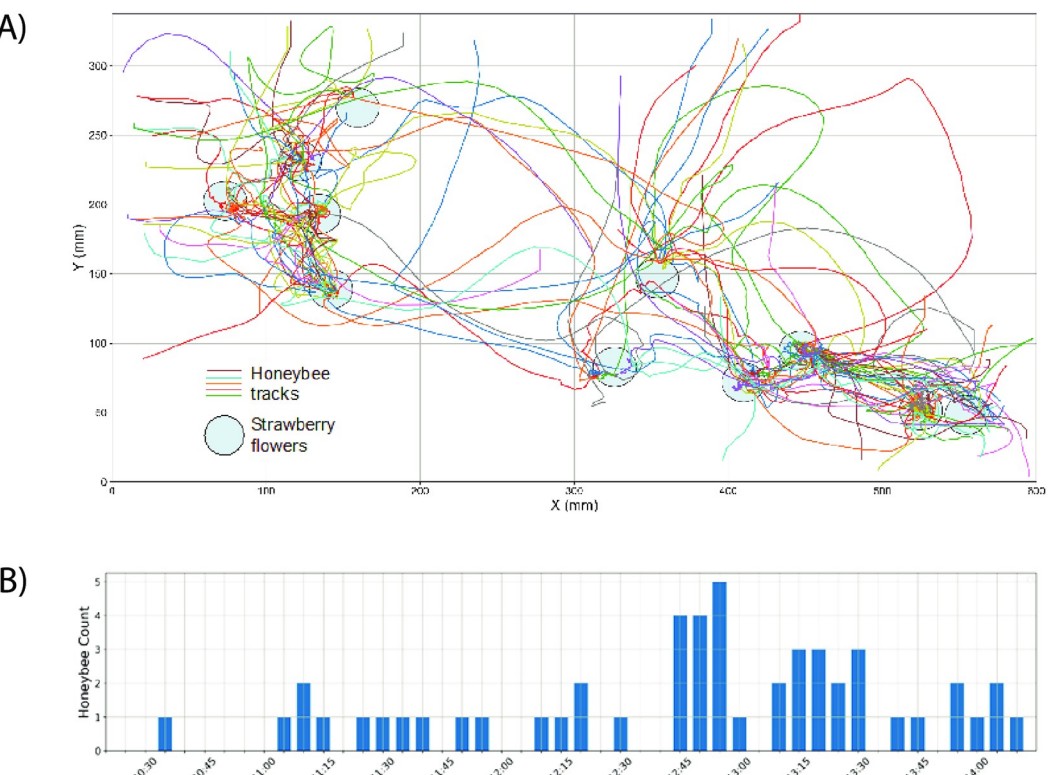

**Fig 7. Honeybee tracking data visualisation.** (A) Honeybee tracks ($n$ = 51) extracted using the automated software and strawberry flowers ($n$ = 11) within the field of view of the camera. Each coloured line represents an individual honeybee track. (B) Count and time of day of honeybee sightings.

## Camera observations

Camera observations recorded 51 honeybees and 20 other insects. Fig 7 shows honeybee tracks extracted from video recordings using the automated software and the time of day of the honeybee sightings.

Out of the recoded 51 honeybees, 45 visited flowers during their foraging bout and out of the recorded 20 non-honeybee insects, only 8 visited flowers. Details on the durations of time spent within the field of view of the camera for all the pollinators are shown in Table 3. Of the total time spent by honeybees in the frame of the camera, 84.40% was foraging on flowers. Of the total time other insects spent in the frame, only 4.60% of recorded time was spent on flowers. Honeybees were only ever recorded landing on flowers. Other insects were recorded landing on flowers, leaves, and strawberry stems.

The field of view of the camera contained 11 open strawberry flowers. Each honeybee visited on average 1.9 flowers, while other insects visited 0.8 flowers on average. The mean number of flowers visited by a honeybee was significantly different from the mean number of flowers visited by other insects ($\chi^2$ = 8.5818, $df$ = 1, $P$ = 0.0034).

We also compared the mean durations that each insect type spent on a flower. Honeybees and other insects spent an average of 8.3 and 14.4 seconds on a flower respectively (medians 3.93 and 5.55 seconds respectively, Fig 8B). A comparison of means showed no significant difference between mean time spent on flowers of honeybees and other insects (W = 114; P = 0.1036).

**Table 3. Distribution of the durations of time spent within the field of view of the camera for all insects.**

| | Time spent (seconds) | | |
|---|---|---|---|
| Insect type | On flower | Not on flower | Total |
| Honeybees | 794.5 | 146.4 | 940.9 |
| Other Insects | 297.9 | 6200.4 | 6498.3 |

Data for honeybees were extracted using automated bee-tracking software and data for other insects were manually extracted from the video recordings.

**Results of the validation study.** Table 4 presents the results of the study to validate honeybee foraging data collected with the automated software. The mean difference between durations measured by software and observed manually was 0.29 ± 0.0628 (*mean ± S. E. M*) seconds.

## Comparing collection methods: Quadrat observations vs. pan-trap counts

In the strawberry polytunnels, collection method had a significant effect on honeybee ($z$ = -3.361; $P < 0.001$) and non-honeybee insect ($z$ = 20.043; $P < 0.001$) capture/observation (Table 5). More honeybees were observed than captured in pan-traps and more other insects were captured in pan-traps than observed.

No honeybees were caught in the raspberry polytunnels with pan-traps, so this comparison was not analysed. Furthermore, collection method did not significantly impact the numbers of non-honeybee insects captured/observed ($z$ = 0.755; $P = 0.451$) in the raspberry crop (Table 5).

In the weed site, collection method significantly impacted the number of honeybees ($z$ = -5.833; $P < 0.001$) and the numbers of other insects ($z$ = 25.263; $P < 0.001$) caught/observed (Table 5). More honeybees were observed than caught in pan-traps and more non-honeybee insects were caught in pan-traps than observed.

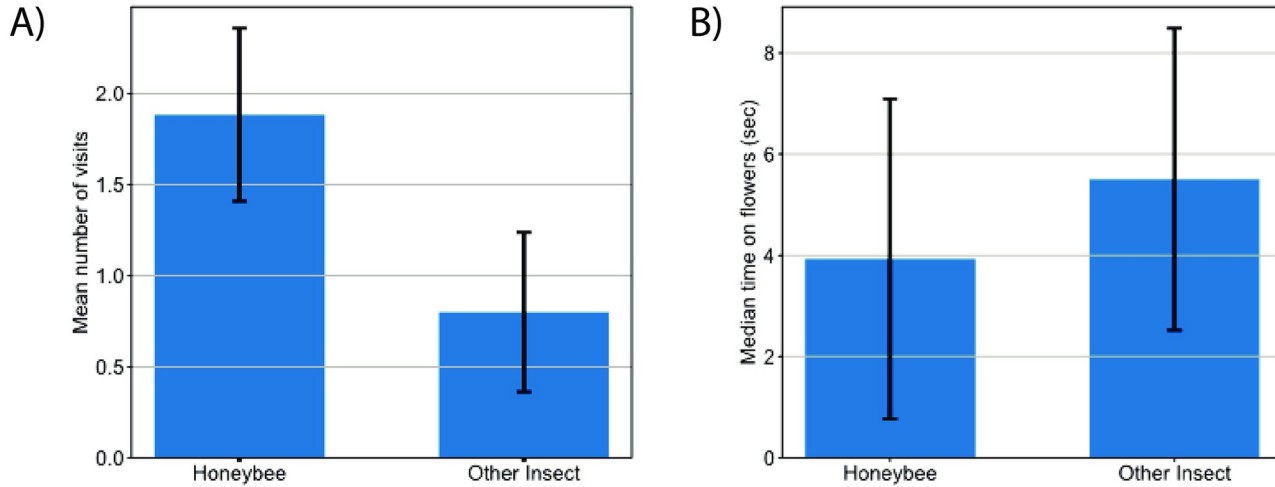

**Fig 8. Video-monitored insect flower visitation summary data.** (A) Mean number of flowers visited by each insect type (per insect) modelled with a negative binomial model. Bars represent the predicted mean number of visits by each pollinator type (per insect) and error bars at 95% confidence intervals for the predicted mean. (B) Median time spent on flowers for each insect type per insect. Error bars indicate the median absolute deviation.

**Table 4. Results of the automated software validation study.**

| Track No. | Time on flowers (seconds) | | Abs. Error (seconds) | Percentage Error |
|---|---|---|---|---|
| | Software | Manual | | |
| 1 | 8.70 | 8.57 | 0.13 | 1.56% |
| 3 | 30.40 | 30.07 | 0.33 | 1.11% |
| 6 | 12.13 | 11.97 | 0.17 | 1.39% |
| 10 | 14.63 | 14.07 | 0.57 | 4.03% |
| 12 | 6.47 | 6.00 | 0.47 | 7.78% |
| 18 | 11.43 | 11.33 | 0.10 | 0.88% |
| 21 | 4.87 | 4.60 | 0.27 | 5.80% |
| 24 | 3.60 | 3.50 | 0.10 | 2.86% |
| 32 | 35.60 | 34.97 | 0.63 | 1.81% |
| 33 | 8.53 | 8.37 | 0.17 | 1.99% |
| **Total** | **136.37** | **133.43** | **2.93** | **2.20%** |

*Track No.* is the number (occurrence sequence) of the recorded honeybee track, *Time on flowers* is the total duration a honeybee spends on strawberry flowers measured using the software and observed from the video manually.

**Table 5. Average insect count per location by pan-trap and by quadrat observation using only the middle of polytunnels and weed area with standard error of the mean (SEM) shown in brackets.**

| | Strawberry polytunnels | | Raspberry polytunnels | | Weed area | |
|---|---|---|---|---|---|---|
| | Honeybees | Other insects | Honeybees | Other insects | Honeybees | Other insects |
| **Quadrats** | 0.33 (0.09) | 2.2 (0.37) | 17.89 (2.26) | 4.22 (1.37) | 29.83 (8.40) | 11.67 (2.88) |
| **Pan-traps** | 0.00 (0.00) | 7.42 (0.96) | 0.00 (0.00) | 3.50 (1.76) | 1.00 (0.58) | 271.33 (20.92) |

## Discussion

### Flower type, location and the impact of sampling method on insect counts

We found flower type to be a factor impacting insect counts of honeybees and non-honeybee insects in our pan-traps and quadrat observations at the agricultural site (Table 5, summary data, Fig 6). The possible interactions and/or relationships between the sampling methods, sampling locations, and insect type are complex. In general, quadrat observations showed weeds to be highly favoured over both crops by both honeybees and other insects. Between the respective crops, we observed many more honeybees in raspberry crops than strawberry crops, but the two crops were similar with respect to other insect observations (Table 1; Fig 6). The pan-traps in the outdoor weeds collected many non-honeybee insects, exceeding the numbers of these captured in both crops by an order of magnitude (Fig 6). When it comes to honeybees though, very few were caught with pan-traps in any area, the method is apparently unsuited to collecting this insect; for example, even though we observed 179 honeybees during three-minute quadrat surveys in the weeds, we caught just three honeybees using pan-traps placed out for relatively extensive 24 hour periods. Other studies similarly demonstrate that pan-trapping is ineffective for some bee species [34,36,41] and that they should either be combined with active insect counting measures, such as sweep netting and/or observation [34], or not used as a primary method for collecting data on bee abundance.

The location and type of sampling within the strawberry and raspberry crop polytunnels significantly influenced insect counts (Tables 1 and 2; Fig 6). For instance, in some cases quadrat observations (Table 1, Fig 6) showed a greater number of honeybees at the edges, compared

to the middle of the strawberry polytunnels. By contrast, in the raspberry polytunnels more non-honeybee insects were observed in the polytunnel middle than at its edge.

## Implications of quadrat observation and pan-trap data for site pollination

On the commercial site, despite the managed hives (Figs 1 and 5), we found honeybees were seldom discovered in strawberry polytunnels compared to raspberry polytunnels and weeds (Fig 6). This was especially evident in quadrat observations. Other studies have made similar findings [25]. Bee pollination is known to improve the yield, quality, shelf life, and commercial value of strawberries [18–21]. However, honeybees do not tend to visit strawberries when other options are available [25], which may be one reason honeybees were found 54, or 90, times less frequently in strawberry polytunnels compared to raspberry, or weed locations (i.e., preferable options were available).

Honeybees and stingless bees have previously been found in higher numbers and making more flower visits at polytunnel edges than in the polytunnel middle [30]. In the current study, the strawberry honeybee observations were also significantly fewer in the polytunnel middle compared to the edge. Pan-traps showed the same result for non-honeybee insects in the strawberry polytunnels. However, observations in the raspberry crop showed an increase in the numbers of non-honeybee insects in the middle compared to the edge. This could be a result of the different lengths of polytunnels, the different crops, or perhaps more interestingly, it could be a result of the differences in insect species foraging behaviours. Overall, the honeybee-related results from the strawberry polytunnels could be due to the extra flight effort required to reach the middle, inaccessibility generally (e.g. due to navigational issues), and/or micro-climatic factors [30]. The non-honeybee insects may simply have spent their entire day (and/or night) within the warm sheltered polytunnels, having no need to return to hives, as there are no eusocial hive-nesting bees in Victoria, besides the honeybee. Variation in honeybee numbers at different polytunnel locations could also be due to some unknown differences between the roads, strips of remnant native vegetation and weeds running around the polytunnel perimeter.

Taken in isolation, results from the quadrat observations and pan-traps indicate that with respect to their numbers, honeybees may not be relatively effective pollinators of the strawberry polytunnel crop. Honeybees are certainly present on the site in large numbers, we have the evidence for this (see Fig 1 hive locations for instance), but they just didn't appear abundant on the strawberry crop through any of the sampling approaches we adopted. Previous work suggests that honeybees are not very attracted to strawberry flowers and thus more bees are needed in the vicinity for successful pollination to occur [22–24] compared to raspberry flowers [22].

Other insects found via quadrat observation included a large number of Diptera and Lepidoptera. It is difficult to fully identify these in Australia [36] and was beyond our scope. Thus, strawberry pollination on the farm may be driven by these insects rather than the honeybees. Often the contribution of non-honeybees/non-bee pollinators is overlooked, especially on commercial farms, however they are of importance to global crop pollination [2]. Globally, non-bee insects are estimated to perform 25–50% of total flower visits and fruit set tends to increase with non-bee insect visitation [2,8]; more precise data on these aspects of pollination from different locations would be helpful. In other countries, stingless bees can pollinate strawberry crops [9,42]. In northern Australia, native stingless bees are also known to be crop pollinators [9,10,43–46]. Despite this, in the southern state of Victoria, Australia there are no eusocial stingless bees as this region has a temperate climate. Promoting and actively supporting wild non-honeybee insects on commercial farms may be key to pollinating crops less

preferred by honeybees, and thus increase crop quality and yield. However, our camera observations need to be taken into account before we rule out the value of honeybees on strawberries completely.

## Camera data provides a contrary pollination story to quadrat observations and pan-traps

Camera observations can record pollinator activity in high resolution [47] enabling temporal analysis impossible with other data collection methods. In our study, pan-traps and quadrat observations certainly recorded higher non-honeybee wild insect counts in strawberry crops than honeybees, but sheer numbers aren't the whole story. Our temporal analysis of camera observations showed that honeybees spend more time when in the field of view of the camera on strawberry flowers than the other insects we recorded (Table 3). This indicates that honeybees may in fact be more effective and efficient pollinators [48] of strawberries than the other insects present at the site, even if they are potentially less numerous.

Video or high resolution still image observations captured with activity-sensed cameras are coming into their own as prices fall and camera resolution, quality, data storage capacity, and speeds improve. They have several advantages over human visual observations including that the footage is stored, accurately time-coded and geo-tagged for later retrieval and detailed analysis. Digital cameras collect data in an objective and fixed region, and their sampling method may be duplicated across time and space. Also, with improvements in image processing and object classification algorithms, comes the potential to automate aspects of ecological video analysis to a high level [47]. When these attributes are compared to the obvious issues of human visual attention, fatigue and expense, camera data has an advantage. The limitations of this study with regard to camera distribution and sampling time are clear. In the future it is conceivable, even desirable, that cameras could be distributed across polytunnel spaces in the same locations and for the same lengths of time, as pan-traps. Only such multi-method sampling approaches can provide the detail needed to unravel the complexities of insect-plant interactions required to provide precision pollination.

Our results showed that insect counts and observations on a commercial strawberry and raspberry farm are influenced by flower type and distance from polytunnel edges. Our results also support existing knowledge that pan-traps are ineffective for collecting honeybee data, but that they are suited to many other insects. Sampling method certainly impacted the type and number of insects on which data could be collected, and also differently informed our understanding of insect-plant interactions on the site. Video capture and data analysis in particular, proved a valuable sampling method in concert with traditional approaches to elucidate pollination interactions. As this technology becomes more accessible, networks of cameras could be placed on agricultural sites to measure and track the numbers and pollination effectiveness of different insects, even in real-time. Such an approach would help us to understand commercial pollination in environments where humans, infrastructure, plants and insects interact in complex ways.

## Acknowledgments

We thank Sunny Ridge Australia for the opportunity to conduct research at their farm.

## Author Contributions

**Conceptualization:** Scarlett R. Howard, Malika Nisal Ratnayake, Adrian G. Dyer, Jair E. Garcia, Alan Dorin.

**Data curation:** Scarlett R. Howard, Malika Nisal Ratnayake, Jair E. Garcia.

**Formal analysis:** Scarlett R. Howard, Malika Nisal Ratnayake, Jair E. Garcia.

**Funding acquisition:** Scarlett R. Howard, Adrian G. Dyer, Alan Dorin.

**Investigation:** Scarlett R. Howard, Malika Nisal Ratnayake, Jair E. Garcia, Alan Dorin.

**Methodology:** Scarlett R. Howard, Malika Nisal Ratnayake, Adrian G. Dyer, Alan Dorin.

**Project administration:** Adrian G. Dyer, Alan Dorin.

**Resources:** Adrian G. Dyer, Alan Dorin.

**Software:** Malika Nisal Ratnayake.

**Supervision:** Adrian G. Dyer, Alan Dorin.

**Validation:** Scarlett R. Howard, Malika Nisal Ratnayake, Jair E. Garcia.

**Writing – original draft:** Scarlett R. Howard.

**Writing – review & editing:** Scarlett R. Howard, Malika Nisal Ratnayake, Adrian G. Dyer, Jair E. Garcia, Alan Dorin.

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
