## [Decision Letter · Decision Letter 0]

1 Mar 2021

PONE-D-20-38002

Towards precision apiculture: traditional and technological insect monitoring methods in strawberry and raspberry crop polytunnels tell different pollination stories

PLOS ONE

Dear Dr. Howard,

Thank you for submitting your manuscript to PLOS ONE. After careful consideration, we feel that it has merit but does not fully meet PLOS ONE’s publication criteria as it currently stands. Therefore, we invite you to submit a revised version of the manuscript that addresses the points raised during the review process.

We look forward to receiving your revised manuscript.

Kind regards,

Guy Smagghe, PhD

Academic Editor

PLOS ONE

Journal Requirements:

2. Thank you for stating the following in the Competing Interests/Financial Disclosure * (delete as necessary) section:

"SRH acknowledges Monash University and the Alfred Deakin Postdoctoral Fellowship. AGD and AD were supported by the Australian Research Council Discovery Projects grant DP160100161. AD acknowledges the support of a Monash-Bosch AgTech Launchpad primer grant for this research. The funders had no role in study design, data collection and analysis, decision to publish, or preparation of the manuscript."

We note that you received funding from a commercial source: Monash-Bosch AgTech Launchpad.

4. We note that Figures 1 and 5 in your submission contain satellite images which may be copyrighted. All PLOS content is published under the Creative Commons Attribution License (CC BY 4.0), which means that the manuscript, images, and Supporting Information files will be freely available online, and any third party is permitted to access, download, copy, distribute, and use these materials in any way, even commercially, with proper attribution. For these reasons, we cannot publish previously copyrighted maps or satellite images created using proprietary data, such as Google software (Google Maps, Street View, and Earth). For more information, see our copyright guidelines: http://journals.plos.org/plosone/s/licenses-and-copyright.

4.1.    You may seek permission from the original copyright holder of Figures 1 and 5 to publish the content specifically under the CC BY 4.0 license. 

4.2.    If you are unable to obtain permission from the original copyright holder to publish these figures under the CC BY 4.0 license or if the copyright holder’s requirements are incompatible with the CC BY 4.0 license, please either i) remove the figure or ii) supply a replacement figure that complies with the CC BY 4.0 license. Please check copyright information on all replacement figures and update the figure caption with source information. If applicable, please specify in the figure caption text when a figure is similar but not identical to the original image and is therefore for illustrative purposes only.

Reviewers' comments:

Reviewer's Responses to Questions

**Comments to the Author**

1. Is the manuscript technically sound, and do the data support the conclusions?

Reviewer #1: Partly

2. Has the statistical analysis been performed appropriately and rigorously? 

Reviewer #1: Yes

3. Have the authors made all data underlying the findings in their manuscript fully available?

Reviewer #1: Yes

4. Is the manuscript presented in an intelligible fashion and written in standard English?

Reviewer #1: Yes

5. Review Comments to the Author

Reviewer #1: The manuscript “Towards precision apiculture: traditional and technological insect monitoring methods in strawberry and raspberry crop polytunnels tell different pollination stories” use three sampling method to monitor insect visitation on crops. In my opinion the most significant contribution of this paper is the demonstration of the usefulness of technology for monitoring pollinators on crops. In general, it is an interesting study, however I have some doubts and suggestions, and there is one inconsistency, all can be found bellow.

The discussion is the part that most needs improvements. Although the division of the discussion into the same sections as the results makes sense, it made the discussion too fragmented. You will see that I made questions and comments pointing out some things that should be discussed and later I saw that they were, but you lost the connection by separating them. It also made the discussion weaker. You should discuss more or inform on the introduction about which behavioral traits make an insect a effective pollinator of the crops you studied.

Line 50 - 52:

Although most alternative insect pollinators are most frequently not managed, stingless bees are, including in Australia. I suggest you include this information.

Line 83:

I think that capture rate does not reflect the data collection from method 1 and 3. Suggestion: visitation rate. Maybe density could also be used. On line 234 you use the word counts, it is also better than capture rate, see also line 306. Even for the pan-trap capture it is not usually called capture rate, even that it is number of individuals/time.

I found it confusing when I read it on the abstract.

Line 160: Reading here I thought “Did you compared the by number of flower visits and time spent on flowers acquired by the software and by human observations? I wonder if differences between honeybees and other insects could be from the method.” I suggest that the validation study should be in the end of this paragraph, on line 163.

Line 279 and table 5: why compare the total of time for all bees and other insects? The number of observed individuals of each class is different, so the mean time spent on flowers and number of flowers visited reflect more the behavior than total time. I suggest that this is removed. The information “Honeybees were only ever observed to land on flowers, whereas other insects landed on flowers, leaves and strawberry stems” is ok.

Line 326: “raspberries saw many more honeybee observations”. I suggest that this is changed to something like “raspberries received many more honeybee visits than…”

Line 337: what is the implication of this in terms of crop pollination? This part of discussion seems more like results. Why is the discussion about this result is on line 351? I suggest that the implications go together with the result.

Line 346 - 348: I don’t see the connection between the first and the last part of the sentence.

Line 348: I don’t understand why or 90 and or weed are between parenthesis, it is confusing, just include them in the sentence.

Line 361: It is important to give information about which are the non-honeybee insects in the results. When you say that these insects may have no need to return to hives, do you mean that they are other social bees? If they are beetles or flies, that doesn’t apply.

Line 366: is abundance a reliable measure of effectiveness of a pollinator? How many visits a strawberry flower needs to be effectively pollinated? I don’t disagree with what you wrote, but I think it should be better discussed (can be combined to information on line 389).

Line 371 - 382: what about stingless bees? They are some papers showing that they are efficient pollinators of strawberries. Although those studies are from Brazil, some in plastic houses and tunnels, Australia has managed native stingless bees and at least Tetragonula carbonaria is used for managed pollination. Could you suggest that this bee group is tested for strawberry pollination in Australia?

Line 385: which kind of spatiotemporal analysis is impossible with direct observation by human eye?

Line 389 - The information on the results is the opposite and the statistical analysis showed no difference on time spent on flowers (lines 291 - 293; figure 8). Fix it on the abstract as well. The mean number of visits is greater for honeybees, but median time on flowers is greater for other insect, according to your text and figure 8.

Line 392: I agree that images/videos have many advantages, but the points you brought here are not entirely true. The variation on counts related to time of day and weather condition can be overcome by making counts on different times of day and weather conditions. Usually it is not done only once a day and is done under good weather conditions. Furthermore, in the case of insects, I don’t think that the presence of the observer may interfere with visitation. For vertebrate, yes, but insects doesn’t seem to care if you are there. I have even touched bees while they were foraging and they just kept doing whatever they were doing. Sometimes they lift a leg, like saying leave me alone, but kept foraging. I suggest that you remove this paragraph and keep the next (lines 398 - 411).

6. PLOS authors have the option to publish the peer review history of their article (what does this mean?). If published, this will include your full peer review and any attached files.

Reviewer #1: **Yes: **Patricia Nunes Silva

---

## [Author Response · Author response to Decision Letter 0]

18 Mar 2021

Editor comment:

We note that Figures 1 and 5 in your submission contain satellite images which may be copyrighted. All PLOS content is published under the Creative Commons Attribution License (CC BY 4.0), which means that the manuscript, images, and Supporting Information files will be freely available online, and any third party is permitted to access, download, copy, distribute, and use these materials in any way, even commercially, with proper attribution. For these reasons, we cannot publish previously copyrighted maps or satellite images created using proprietary data, such as Google software (Google Maps, Street View, and Earth). For more information, see our copyright guidelines: http://journals.plos.org/plosone/s/licenses-and-copyright.

We require you to either (1) present written permission from the copyright holder to publish these figures specifically under the CC BY 4.0 license, or (2) remove the figures from your submission.

Author response:

Thank you for this information. The author team has now created new diagrams for Figures 1 and 5 that can be published 

under the Creative Commons Attribution License (CC BY 4.0).

Reviewer comment 1:

Reviewer #1: The manuscript “Towards precision apiculture: traditional and technological insect monitoring methods in strawberry and raspberry crop polytunnels tell different pollination stories” use three sampling method to monitor insect visitation on crops. In my opinion the most significant contribution of this paper is the demonstration of the usefulness of technology for monitoring pollinators on crops. In general, it is an interesting study, however I have some doubts and suggestions, and there is one inconsistency, all can be found bellow.

The discussion is the part that most needs improvements. Although the division of the discussion into the same sections as the results makes sense, it made the discussion too fragmented. You will see that I made questions and comments pointing out some things that should be discussed and later I saw that they were, but you lost the connection by separating them. It also made the discussion weaker. You should discuss more or inform on the introduction about which behavioral traits make an insect a effective pollinator of the crops you studied.

Author response 1:

 We thank the reviewer for their supportive comments and suggestions which have improved the quality of our manuscript.

Below we detail where we have revised the manuscript based on the reviewer’s suggestions, particularly in the discussion as they have indicated. All referenced line numbers refer specifically to the manuscript with accepted track changes (rather than the one highlighting the changes we made using Word’s “track changes” feature).

Reviewer comment 2:

Line 50 - 52:

Although most alternative insect pollinators are most frequently not managed, stingless bees are, including in Australia. I suggest you include this information.

Author response 2:

 We thank the reviewer for the suggestion. We have now added more detail and citations for this point in lines 54-56: “In Australia, there are some exceptions to this including stingless bees [9] such as Tetragonula carbonaria [10], and other native bee genera and families, including Xylocopa, Amegilla, and Megachilidae spp. [11-17].” 

Reviewer comment 3:

Line 83:

I think that capture rate does not reflect the data collection from method 1 and 3. Suggestion: visitation rate. Maybe density could also be used. On line 234 you use the word counts, it is also better than capture rate, see also line 306. Even for the pan-trap capture it is not usually called capture rate, even that it is number of individuals/time.

I found it confusing when I read it on the abstract.

Author response 3:

 Thank you for the suggestion. All places where we had written ‘capture rates’ have now been changed throughout the manuscript with “insect counts” or “counts”

Reviewer comment 4:

Line 160: Reading here I thought “Did you compared the by number of flower visits and time spent on flowers acquired by the software and by human observations? I wonder if differences between honeybees and other insects could be from the method.” I suggest that the validation study should be in the end of this paragraph, on line 163.

Author response 4:

The reviewer is correct in that we did compare the software analysis to human analysis of the video data. This is described in the Validation Experiment Methods and Results section. To address the need for clarification raised by reviewer 1, we have also indicated that this was done in lines 172-173: “A validation experiment comparing human observations of the video footage and software analysis is described below.”

Reviewer comment 5:

Line 279 and table 5: why compare the total of time for all bees and other insects? The number of observed individuals of each class is different, so the mean time spent on flowers and number of flowers visited reflect more the behavior than total time. I suggest that this is removed. The information “Honeybees were only ever observed to land on flowers, whereas other insects landed on flowers, leaves and strawberry stems” is ok.

Author comment 5:

 We have now re-written it to clarify: “Of the total time spent by honeybees in the frame of the camera, 84.4 % was foraging on flowers. Of the total time other insects spent in the frame, only 4.60 % of recorded time was spent on flowers. Honeybees were only ever recorded landing on flowers. Other insects were recorded landing on flowers, leaves and strawberry stems.” See lines 287-291.

Reviewer comment 6:

Line 326: “raspberries saw many more honeybee observations”. I suggest that this is changed to something like “raspberries received many more honeybee visits than…”

Author response 6:

 We have now clarified this sentence in lines 339-342: “Between the respective crops, we observed many more honeybees in raspberry crops than strawberry crops, but the two crops were similar with respect to other insect observations (Table 1; Fig. 6).”

Reviewer comment 7:

Line 337: what is the implication of this in terms of crop pollination? This part of discussion seems more like results. Why is the discussion about this result is on line 351? I suggest that the implications go together with the result.

Author response 7:

Here we are summarising the results for discussion purposes. To make this clear and direct the reader to the results we are referring to, we have now included references to Tables 1 & 2 and Figure 6.

Reviewer comment 8:

Line 346 - 348: I don’t see the connection between the first and the last part of the sentence.

Author response 8:

We have now reviewed this section and agree with the reviewer comment. We have therefore divided the sentence from the initial submission into two single sentences in lines 362-366. This clarifies the message: “Bee pollination is known to improve the yield, quality, shelf life, and commercial value of strawberries [18-21]. However, honeybees do not tend to visit strawberries when other options are available [25], which may be one reason honeybees were found 54, or 90, times less frequently in strawberry polytunnels compared to raspberry, or weed locations, respectively, (i.e., preferable options were available).”

Reviewer comment 9:

Line 348: I don’t understand why or 90 and or weed are between parenthesis, it is confusing, just include them in the sentence.

Author response 9:

We have now changed this as per the reviewer suggestion (lines 363-366): “However, honeybees do not tend to visit strawberries when other options are available [25], which may be one reason honeybees were found 54, or 90, times less frequently in strawberry polytunnels compared to raspberry, or weed locations (i.e., preferable options were available).”

Reviewer comment 10:

Line 361: It is important to give information about which are the non-honeybee insects in the results. When you say that these insects may have no need to return to hives, do you mean that they are other social bees? If they are beetles or flies, that doesn’t apply.

Author response 10:

They are non-honeybees and the assumption is that they are not social insects. There are no native eusocial bees in Victoria (they are restricted to northern Australia) and bumblebees are only present in Tasmania (not on the mainland of Australia). So, it is possible to conclude that besides the honeybee, the other insects were not social. 

We have clarified this in lines 377-380: “The non-honeybee insects may simply have spent their entire day (and/or night) within the warm sheltered polytunnels, having no need to return to hives, as there are no eusocial hive-nesting bees in Victoria, besides the honeybee.”

Reviewer comment 11:

Line 366: is abundance a reliable measure of effectiveness of a pollinator? How many visits a strawberry flower needs to be effectively pollinated? I don’t disagree with what you wrote, but I think it should be better discussed (can be combined to information on line 389).

Author response 11:

Strawberry flowers are not particularly attractive to honeybees and so, for successful pollination, more hives in strawberry fields are required than raspberry fields. This is discussed in the introduction: “Between 6 and 16 visits from honeybees are required for full pollination of a strawberry flower [22-24] and 12 – 25 honeybee hives per hectare are advised [22, 24-26]. It has previously been reported that honeybees are often not very attracted to strawberry nectar and pollen [22, 24, 25]. Conversely, raspberry (Rubus sp.) flowers appear to be attractive to honeybees due to a relatively high quantity of nectar [24, 27]. Since raspberry flowers are attractive, fewer hives are needed for pollination of these flowers when compared to strawberries, with data indicating only between 0.5 and 2.5 hives per hectare are required for raspberries [22].” In lines 60-67.

We have now also added this information in the discussion, lines 388-390: “Previous work suggests that honeybees are not very attracted to strawberry flowers and thus more bees are needed in the vicinity for successful pollination to occur [22-24] compared to raspberry flowers [22].”

Reviewer comment 12:

Line 371 - 382: what about stingless bees? They are some papers showing that they are efficient pollinators of strawberries. Although those studies are from Brazil, some in plastic houses and tunnels, Australia has managed native stingless bees and at least Tetragonula carbonaria is used for managed pollination. Could you suggest that this bee group is tested for strawberry pollination in Australia?

Author response 12:

While non-Australian stingless bee species are known to pollinate strawberries, in Victoria, Australia (where our study took place) there are no eusocial stingless bees.

In lines 398-401, we have included a couple of sentences detailing the current and future value of Australian stingless bees: “In other countries, stingless bees can pollinate strawberry crops [9, 42]. In northern Australia, native stingless bees are also known to be crop pollinators [9, 10, 43-46]. Despite this, in the southern state of Victoria, Australia there are no eusocial stingless bees as this region has a temperate climate.”

Reviewer comment 13:

Line 385: which kind of spatiotemporal analysis is impossible with direct observation by human eye?

Author response 13:

It is now well-acknowledged that the human eye is limited by attention and it is most frequently not possible to reliably estimate greater than one active event simultaneously (see Simons, D.J. and Chabris, C.F., 1999. Gorillas in our midst: Sustained inattentional blindness for dynamic events. Perception, 28(9), pp.1059-1074.) 

We have removed ‘spatio’ from the manuscript to make this point clearer to readers.

Reviewer comment 14:

Line 389 - The information on the results is the opposite and the statistical analysis showed no difference on time spent on flowers (lines 291 - 293; figure 8). Fix it on the abstract as well. The mean number of visits is greater for honeybees, but median time on flowers is greater for other insect, according to your text and figure 8.

Author response 14:

We apologise for the confusion. There are two different measurements in the paper on insect flower visitation showing different results. 

Result 1: Of the time honeybees spent in the field of view of the camera, 84.4 % of that time was spent only on flowers. For other insects, of time spent in the field of view of the camera, only 4.60 % of that time was spent on flowers. This is written in lines 287-291 and as seen in Table 5. 

Result 2: We also recorded the mean time spent on a flower for honeybees and other insects. This is the result which is not significantly different as honeybees and other insects spent an average of 8.3 and 14.4 seconds on a flower respectively (Figure 8b).

In the statement from the reviewer we are referring to the first result and thus we have edited the sentence to clarify this point in lines 411-413: “Our temporal analysis of camera observations showed that honeybees spend more time when in the field of view of the camera on strawberry flowers than the other insects we recorded (Table 5).”

We have also changed the abstract (lines 33-34): “Although honeybees were relatively scarce among strawberry crops, camera data shows they spent more time visiting flowers than other insects.”

Reviewer comment 15:

Line 392: I agree that images/videos have many advantages, but the points you brought here are not entirely true. The variation on counts related to time of day and weather condition can be overcome by making counts on different times of day and weather conditions. Usually it is not done only once a day and is done under good weather conditions. Furthermore, in the case of insects, I don’t think that the presence of the observer may interfere with visitation. For vertebrate, yes, but insects doesn’t seem to care if you are there. I have even touched bees while they were foraging and they just kept doing whatever they were doing. Sometimes they lift a leg, like saying leave me alone, but kept foraging. I suggest that you remove this paragraph and keep the next (lines 398 - 411).

Author response 15:

The reviewer is correct, and we have thus deleted the paragraph.

---

## [Decision Letter · Decision Letter 1]

29 Apr 2021

Towards precision apiculture: traditional and technological insect monitoring methods in strawberry and raspberry crop polytunnels tell different pollination stories

PONE-D-20-38002R1

Dear Dr. Howard,

We’re pleased to inform you that your manuscript has been judged scientifically suitable for publication and will be formally accepted for publication once it meets all outstanding technical requirements.

Kind regards,

Guy Smagghe, PhD

Academic Editor

PLOS ONE

Additional Editor Comments (optional):

Reviewers' comments:

Reviewer's Responses to Questions

**Comments to the Author**

1. If the authors have adequately addressed your comments raised in a previous round of review and you feel that this manuscript is now acceptable for publication, you may indicate that here to bypass the “Comments to the Author” section, enter your conflict of interest statement in the “Confidential to Editor” section, and submit your "Accept" recommendation.

Reviewer #1: All comments have been addressed

2. Is the manuscript technically sound, and do the data support the conclusions?

Reviewer #1: Yes

3. Has the statistical analysis been performed appropriately and rigorously? 

Reviewer #1: Yes

4. Have the authors made all data underlying the findings in their manuscript fully available?

Reviewer #1: Yes

5. Is the manuscript presented in an intelligible fashion and written in standard English?

Reviewer #1: Yes

6. Review Comments to the Author

Reviewer #1: (No Response)

7. PLOS authors have the option to publish the peer review history of their article (what does this mean?). If published, this will include your full peer review and any attached files.

Reviewer #1: No

---

## [Editor Report · Acceptance letter]

5 May 2021

PONE-D-20-38002R1 

Towards precision apiculture: traditional and technological insect monitoring methods in strawberry and raspberry crop polytunnels tell different pollination stories 

Dear Dr. Howard:

I'm pleased to inform you that your manuscript has been deemed suitable for publication in PLOS ONE. Congratulations! Your manuscript is now with our production department. 

Kind regards, 

on behalf of

Prof. Guy Smagghe 

Academic Editor

PLOS ONE